# Evaluating the Removal of the Antibiotic Cephalexin from Aqueous Solutions Using an Adsorbent Obtained from Palm Oil Fiber

**DOI:** 10.3390/molecules26113340

**Published:** 2021-06-02

**Authors:** Nancy Acelas, Sandra M. Lopera, Jazmín Porras, Ricardo A. Torres-Palma

**Affiliations:** 1Grupo de Materiales con Impacto, MAT&MPAC, Facultad de Ciencias Básicas, Universidad de Medellín, Medellín 050010, Colombia; 2Grupo de Investigación en Remediación Ambiental y Biocatálisis (GIRAB), Instituto de Química, Facultad de Ciencias Exactas y Naturales, Universidad de Antioquia UdeA, Calle 70 No. 52-21, Medellín 050010, Colombia; samy2810@gmail.com (S.M.L.); ricardo.torres@udea.edu.co (R.A.T.-P.); 3Grupo de Investigaciones Biomédicas Uniremington, Facultad de Ciencias de la Salud, Corporación Universitaria Remington (Uniremington), Calle 51 No. 51-27, Medellín 050010, Colombia; jazmin.porras@uniremington.edu.co

**Keywords:** adsorption, cephalexin, wastewater, biochar, isotherm, kinetics

## Abstract

This study aimed to understand the adsorption process of cephalexin (CPX) from aqueous solution by a biochar produced from the fiber residue of palm oil. Scanning electron microscopy, Fourier transform infrared spectroscopy, Boehm titration, and the point of zero charge were used to characterize the morphology and surface functional groups of the adsorbent. Batch tests were carried out to evaluate the effects of the solution pH, temperature, and antibiotic structure. The adsorption behavior followed the Langmuir model and pseudo-second-order model with a maximum CPX adsorption capacity of 57.47 mg g^−1^. Tests on the thermodynamic behavior suggested that chemisorption occurs with an activation energy of 91.6 kJ mol^−1^ through a spontaneous endothermic process. Electrostatic interactions and hydrogen bonding represent the most likely adsorption mechanisms, although π–π interactions also appear to contribute. Finally, the CPX removal efficiency of the adsorbent was evaluated for synthetic matrices of municipal wastewater and urine. Promising results were obtained, indicating that this adsorbent can potentially be applied to purifying wastewater that contains trace antibiotics.

## 1. Introduction

Antibiotics are indispensable products for human life, and their consumption has increased to combat the spread of infectious diseases in humans and animals [1]. However, 30–90% of consumed antibiotics are not metabolized in the body and are discharged into aquatic systems as active compounds [2]. Cephalexin (CPX) is an antibiotic of the cephalosporin family, which is the second-most consumed group of antibiotics worldwide [3], and is normally used to treat respiratory and urinary infections [4]. The discharge of contaminated water from pharmaceutical facilities, hospitals, homes, and agro-industrial facilities containing pharmaceuticals such as CPX into the environment can have adverse impacts on human health, increase antibiotic resistance, and inhibit the growth of algae and beneficial microorganisms in the environment [5]. Therefore, it is important to explore efficient and cost-effective technologies to remove antibiotics from polluted water.

Adsorption is a promising and available technique for removing many contaminants in water. For example, biochar is a carbon-rich material produced from biomass that is excellent at removing organic contaminants [2,6,7,8,9,10,11]. Biochar has a porous structure, high aromaticity, and various functionalities that confer the capability for different interactions and coupling with antibiotic molecules [5]. Several oxygenated functional groups on its surface, including alcohols, carboxylic acids, phenols, and ethers, allow interactions such as π–π electron donor–acceptor interactions, electrostatic interactions, and hydrogen bonding that have been established as major adsorption mechanisms between antibiotics and biochar [6].

One of the most popular methods for producing low-cost biochar adsorbents is through pyrolysis and subsequent chemical activation of the biomass waste. This method also helps mitigate issues with biomass waste disposal. Therefore, several agro-industrial residues (e.g., alligator weed [2], walnut shells [7], bamboo waste [1], tea industry wastes [8], *Albizia lebbeck* seed pods [9], pistachio shells [10], alfalfa hay [11], and rice husks [5,12,13]) have been studied for their ability to remove antibiotics and other pharmaceuticals from water. Most of these studies have focused on determining the maximum adsorption capacities of these residues. A wide range of adsorption capacities has been reported that depends on both the type of agro-industrial residue and the preparation procedure [5,12,13]. The use of palm oil residues as adsorbents is of special interest as the palm oil agro-industry generates a significant amount of residual biomass (~87 million t/y) [14]. Several studies have demonstrated that biochar produced from palm oil and other agro-industrial residues is an effective adsorbent for removing heavy metals [15,16,17], dyes [18,19], pharmaceuticals [20,21,22,23], and other organic pollutants [24] from aqueous systems.

Thus, the objective of this study was to produce and apply biochar derived from fiber and shell residues of palm oil to remove CPX from water. Batch experiments were performed to investigate the CPX adsorption behavior of biochar. The effects of the solution pH, temperature, matrix, and antibiotic structure were evaluated. The adsorption kinetics, isotherm, and thermodynamics were evaluated to characterize the adsorption behavior in detail. Finally, structural properties of the adsorbent, including the pore size, volume, and surface functional groups, were analyzed to clarify the CPX adsorption process of biochar produced from palm oil residue.

## 2. Results and Discussion

### 2.1. CPX Removal Performance of Each Adsorbent

The differences in the adsorption properties of F and S were evaluated to determine the influences of the original material and activation process. Batch-type experiments were carried out where each adsorbent was placed in contact with a solution with 20 mg L^−1^ of CPX. As shown in Figure 1, F_Zn_ has the highest CPX removal efficiency at 74.1%, while S_Zn_ and F_HA_ had similar removal efficiencies of 17.5% and 17.4%, respectively.

This behavior can be associated with the nature of the prepared adsorbent and is characterized by the pH at PZC (pH_PZC_), which is a parameter for identifying the variation in the surface charge of the adsorbent with the pH. If pH_PZC_ is known, then the pH values at which the biochar would have a positive or negative charge can be determined. This is useful for determining the pH conditions and chemical structure of the pollutant that would favor the adsorption process. As presented in Table 1, F_Zn_, S_Zn_, and F_HA_ had pH_PZC_ values of 4.0, 2.3, and 4.8, respectively. At the pH of the experiment (pH = 6.5), all of these adsorbents would have a negative surface charge to interact with the positive charge of CPX, which would be in zwitterionic form (pK_a1_ 2.56 and pK_a1_ 6.88). Thus, all adsorbents had the potential for electrostatic interaction. This indicates that pH_PZC_ by itself does not explain the behavior of adsorbents during the CPX adsorption process. Therefore, the morphology and surface area also needed to be considered.

The results with the BET method showed that the specific surface areas of F_Zn_, S_Zn_, and F_HA_ were 835.3, 575.1, and 14.8 m^2^ g^−1^, respectively (see Table 1).

These results indicate that the activating agent is a fundamental key for the modification of biomass to produce an activated carbon. As shown in Figure 2, adsorbents produced with ZnCl_2_ as the activating agent had a more porous structure than the adsorbent produced with H_3_PO_4_. These pores can be partially attributed to the effect of ZnCl_2_, which is a dehydrating reagent that promotes the decomposition of carbonaceous materials, restricts the formation of tars, and induces carbon aromatization [15]. These observations are in agreement with the surface areas from the BET analysis, where F_Zn_ had the largest surface area. This may be directly correlated with a higher CPX removal efficiency.

However, the surface area is not the only determining factor for the CPX adsorption process, because S_Zn_ had a surface area 39 times greater than that of F_HA_, but both adsorbents had similar CPX removal efficiencies (~20%). This indicates that the surface functional groups play an important role in the adsorption process.

Appendix A shows the infrared (IR) spectra for the three adsorbents before and after the CPX adsorption process. Both F_Zn_ and F_HA_ had carboxylic acid and phenol functional groups, as evidenced by the amplitude of the signal at 3600 cm^−1^, corresponding to the –OH functional group. Here, S_Zn_ did not have a large amplitude. After adsorption, the signal at 3600 cm^−1^ became less intense for F_Zn_, indicating that CPX may have interacted with these oxygenated functional groups. Boehm titration was carried out (Table 1) to verify and quantify the acid groups on F_Zn_, which had the highest CPX removal efficiency. The carboxylic acid groups were more abundant than the phenols and lactones by 8.3 and 11 times, respectively.

The above results indicate that both the activating method and biomass type can considerably affect CPX adsorption because they can be varied to produce adsorbents with different physicochemical characteristics such as the morphology, functional groups, and surface area. These results suggest that effects other than the surface area influence the adsorption and that the surface functional groups may be the main factor for CPX removal.

The pH is known as a key factor in adsorption processes, and it influences the chemical nature of the functional groups and contaminant. Thus, the effects of the functional groups and pH on CPX adsorption were evaluated in further detail. F_Zn_ was selected for further analysis because it was the adsorbent with the highest CPX removal efficiency.

### 2.2. Effect of pH on CPX Adsorption

The solution pH influences the surface charge and surface functional groups of the adsorbent as well as the CPX structure (related to pKa). Thus, pH was varied to evaluate the effect on CPX removal, with the results shown in Figure 3. The CPX removal decreased as the pH was increased from 2 to 12. At pH values of 2–6, the CPX removal was around 90%; at pH values of >6.5, the CPX removal decreased to 20% at pH 9.2 and 6% at pH 11.5. In control experiments, hydrolysis of CPX at these pH values was discarded (data not shown).

This relationship between CPX removal and solution pH can be explained by the properties of the adsorbent, such as the surface charge (pH_pzc_~ 4.0) and surface functional groups, and CPX, such as its chemical structure and acid–base properties (pK_a_ values) (Figure 3) [13].

The similar levels of CPX removal from pH 2 to 6 may be related to the fact that the adsorbent had both positive and negative charges at these pH values. At pH 2, the adsorbent was slightly positive, and at pH 6, the adsorbent was slightly negative. Hence, within this pH range, CPX was in the zwitterionic form; it had both positive and negative charges, which favored adsorption by electrostatic interactions. These results indicate that this adsorbent favors the adsorption of charged pollutants. If the pollutant is positively charged, adsorption occurs over a wide pH range, favoring pH < 8. However, if the contaminant is negatively charged, the adsorption decreases when pH > 8. Similar reports [25] have shown that the CPX removal efficiency increases with decreasing pH, which agrees with the results of the present study. This indicates that, in addition to the electrostatic charge, the interaction between surface functional groups is also important for the CPX removal efficiency.

### 2.3. Effect of Chemical Structure on CPX Adsorption

Changes to adsorption due to pH are affected by the structure of the antibiotic, which may improve or worsen the interaction between the adsorbate and adsorbent. The influence of the chemical structure of the pharmaceutical on the removal process was evaluated to clarify its interactions with the adsorbent. As shown in Figure 4, three pharmaceuticals with different chemical structures were tested for their removal by F_Zn_, i.e., naproxen (NPX), acetaminophen (ACE), and cephalexin (CPX).

NPX and CPX followed similar trends with removal efficiencies of ~90% during the first 7 min of treatment, after which adsorption equilibrium was achieved. Meanwhile, ACE had a maximum removal efficiency of around 20%. To explain these results, the pH_PZC_ of F_Zn_ and pK_a_ of the pharmaceuticals were considered. Under natural pH conditions (4.0–5.0), the pK_a_ values indicate that the pharmaceuticals have the following chemical form: CPX is principally zwitterionic, ACE is zero charge (with a protonated phenol group), and NPX is negatively charged. Meanwhile, the adsorbent has a pH_PZC_ between 2.0 and 4.0, so it has negative and positive charges.

Thus, the high removal efficiencies for CPX and NPX can be attributed to electrostatic interactions with the adsorbent that are not present with ACE under the working conditions. The 20% removal efficiency of ACE can be related to other types of interactions such as hydrogen bonds, π–π interactions, or complexation [5], which should also contribute to the removal of CPX and NPX. According to Boehm titration (Table 1), the adsorbent has phenolic groups on its surface, which have the ability to form a complex with the carboxylic unprotonated group of NPX or CPX and through hydrogen bonds with the carbonyl group on ACE. Zeng et al., 2018 [5] found that hydrogen bonding is a dominant mechanism for the adsorption of the antibiotics doxycycline and ciprofloxacin on biochars. For F_Zn_, electrostatic attraction appears to be the most important interaction compared with hydrogen bonds and π–π interactions.

### 2.4. Effect of Adsorbent Dose

Figure 5 shows the effect of the adsorbent dose on the CPX adsorption capacity (*Q_e_*, which is *Q_t_* at the equilibrium) and removal efficiency. The adsorbent dose was varied from 0.2 to 2 g L^−1^ under the favorable conditions previously determined. Increasing the adsorbent dose decreased the CPX adsorption capacity while increasing the CPX removal efficiency. As shown in Figure 5 the CPX adsorption capacity decreased from 50 mg g^−1^ to 10 mg g^−1^, and the removal efficiency increased from 40% to 98%. The increase in the removal efficiency with the adsorbent dose can be attributed to the increased surface area and availability of more adsorption sites [26]. However, when the F_Zn_ dose is low, the fewer adsorption sites limit the increase in the removal efficiency, reducing the CPX removal efficiency [27]. At higher F_Zn_ doses, aggregation limits the number of active surface sites, which slows the increase in the removal efficiency and reduces the adsorption capacity [28].

The optimum adsorbent dose was determined to be 1.6 g L^−1^ because this was the minimum dose with the highest removal efficiency and stable adsorption capacity. Similar results have been reported for CPX removal by other activated carbonaceous materials [2,9,29]. These studies also observed that the removal efficiency rapidly increased and the equilibrium adsorption capacity decreased as the adsorbent dose was increased up to 1.6 g/L, after which both the removal efficiency and adsorption capacity remained unchanged [2,5].

### 2.5. Adsorption Kinetics

Adsorption kinetics can be used to predict the removal efficiency for a contaminant from an aqueous solution. Controlling factors can be determined, such as the rate of chemical reactions and the mass transfer mechanism. In this study, the PFO [30], PSO [31], and intra-particle diffusion [32] models were used to evaluate the adsorption of different concentrations of CPX on F_Zn_. The lineal equations of the PFO and PSO models can be expressed as follows:(1)$$\log\left(q_{e}-q_{t}\right)=\log q_{e}-\frac{k_{1}}{2.303}t$$
(2)$$\frac{t}{q_{t}}=\frac{1}{k_{2}q_{e}{}^{2}}+\frac{t}{q_{e}}$$
where *q_t_* and *q_e_* are the adsorption capacities of CPX at time *t* and at equilibrium, respectively (mg g^−1^); *t* is the reaction time (min); and *k*_1_ and *k*_2_ are the adsorption rate constants of PFO and PSO, respectively (g mg^−1^ min^–1^). *q_e_* and the rate constants of the PFO and PSO models can be determined from the slope and intercept of the plot of log (*q_e_* − *q_t_*) versus *t* and *t*/*q_t_* versus *t*. Appendix A shows the lineal equation adjusted for each model. The intra-particle diffusion rate parameter can be expressed as follows:(3)$$q_{t}=k_{di}\sqrt{t}+C_{i}$$

The intra-particle diffusion rate constant $k_{di}\;$(mg g^−1^ min^−1/2^) and $C_{i}$ can be calculated from the slope and intercept of these plots. For all concentrations in this study, the correlation coefficient (*R*^2^) and normalized standard deviation (Δ*q_e_*, %) given in Table 2 indicated that the PSO model fit the experimental data better than the PFO model did. This indicates that the adsorption process can be attributed to chemical adsorption [10] through electron sharing or transfer between F_Zn_ and CPX. These results are consistent with other studies on the adsorption of pharmaceuticals on carbonaceous materials [6,12,13,33,34]. Appendix A shows the effect of the initial concentration of CPX on the removal efficiency.

The CPX removal efficiency increased remarkably with an increase in the initial concentration. During the initial step, the adsorption rate was high owing to the driving forces from the high concentration and huge number of active sites. The intra-particle diffusion model was used to describe the diffusion of CPX molecules onto F_Zn_. According to Equation (3), if the linear trend line fit to *q_t_* versus *t*^0.5^ passes through the origin point (i.e., C_i_ = 0), the intra-particle diffusion is the only rate-limiting step for the adsorption kinetics. As demonstrated in Table 2, all calculated C_i_ values were greater than zero, indicating that more than one mechanism controlled the adsorption process. The adsorption rate constants for the first adsorption stage (*K*_1*d*_) were higher than those for the second adsorption stage (*K*_2*d*_). This indicates that the external diffusion of CPX molecules from the solution to the adsorbent surface was faster than the diffusion of CPX molecules onto the adsorbent pores. This also indicates that the rate of CPX adsorption was initially faster because of the availability of more active sites at the beginning of the adsorption process.

### 2.6. Isotherm Studies

Adsorption isotherm models describe the distribution of the adsorbed contaminant on the adsorbent after the equilibrium state is reached. The information provided by these models is very important to understand the adsorption process. In this study, the Langmuir [35] and Freundlich [36] models were applied to analyze the experimental data. The Langmuir isotherm assumes monolayer adsorption and is widely used to quantify the maximum adsorption capacity (*Q_m_*):(4)$$\frac{C_{e}}{Q_{e}}=\frac{1}{Q_{m}K_{L}}+\frac{C_{e}}{Q_{m}}$$
where *Q_m_* is the maximum adsorption capacity (mg g^−1^) and *C_e_* and *Q_e_* are the equilibrium CPX concentration (mg L^−1^) and equilibrium adsorption capacity (mg CPX g^−1^ of F_Zn_), respectively. *Q_m_* and the constant *K_L_* (L g^−1^) can be calculated from the slope and intercept (Appendix A). The Langmuir isotherm can be used to determine the degree of affinity between CPX and F_Zn_:(5)$$R_{L}=\frac{1}{1+C_{e}\;K_{L}}$$

The Freundlich isotherm model assumes multilayer adsorption, and it is often used for heterogeneous surfaces. A linear form of the Freundlich equation is given by [37]
(6)$$\log Q_{e}=\log K_{F}+\frac{1}{n}\log C_{e}$$
where *Q_e_* is the equilibrium adsorption capacity (mg g^−1^); *n* (g L^−1^) and *K_F_* are the adsorption intensity (i.e., degree of adsorption favorability) and adsorption capacity, respectively; and *C_e_* is the concentration (mg L^−1^).

Figure 6 shows the experimental and theoretical isotherms of CPX on F_Zn_. The adsorption capacity increased with the equilibrium concentration at 0–12 mg L^−1^. At *R*^2^ = 0.98, the Langmuir model best described the experimental data of CPX adsorption on F_Zn_, which suggests monolayer adsorption (see Appendix A).

The Langmuir separation factor (*R_L_*) in Equation (5) was used to determine if the adsorption process was irreversible (*R_L_* = 0), linear (*R_L_* = 1), favorable (0 < *R_L_* < 1), or unfavorable (*R_L_* > 1) [38]. In this study, *R_L_* = 0.15 was considered to indicate that the CPX adsorption on F_Zn_ was favorable. The maximum monolayer adsorption capacity (*Q_m_*) was 57.47 mg/g, which is comparable to values reported for CPX adsorption with several types of biomass (Table 3).

### 2.7. Thermodynamic Behavior

To evaluate the thermodynamic parameters for CPX adsorption on F_Zn_, experiments were carried out at 15, 25, and 35 °C. Kinetic data obtained by the PSO model such as k_2_ were used with the Arrhenius equation to obtain the activation energy *E_a_* (kJ mol^−1^) [39,40]:(7)$$lnk_{2}=lnA_{0}-\frac{E_{a}}{RT}$$
where *A*_0_ is the Arrhenius constant, *R* is the universal gas constant (8.314 J mol^−1^ K^−1^), and *T* is the temperature (K). *E_a_* is defined as the minimal energy that CPX overcomes for adsorption and was calculated from the slope of the plot ln *k*_2_ versus 1/*T* to be 91.6 kJ mol^−1^ (see Appendix A).

According to previous studies, a low activation energy of 5–50 kJ mol^−1^ suggests physical adsorption, whereas *E_a_* > 50 kJ mol^−1^ implies chemical adsorption. Therefore, the obtained *E_a_* value indicates that chemisorption is the predominant mechanism for CPX adsorption on F_Zn_. This agrees with the fit to the PSO model and with the electrostatic interaction and influence of functional groups, as previously discussed.

The thermodynamic parameters of the standard free energy (Δ*G*°), enthalpy (Δ*H*°), and entropy (Δ*S*°) were determined as follows [39,41]:(8)$$lnk_{c}=\frac{\Delta S}{R}-\frac{\Delta H}{RT}$$
(9)$$\Delta G_{ads}=\Delta H_{ads}-T\Delta S_{ads}$$
where *R* (8.314 J/mol K) is the ideal gas constant, *T* (K) is the temperature, and *k**_c_* (L/g) is the standard thermodynamic equilibrium constant defined by *q*e/*C**e*. Δ*H*° and Δ*S*° can be estimated from the slope and intercept, respectively, for the plot of ln *k**_c_* versus 1/*T* (see Appendix A).

Table 4 demonstrates that Δ*G*° became more negative as the temperature increased. This indicates that CPX adsorption is a spontaneous process favored by increasing temperature. The positive Δ*S*° confirmed the increased randomness at the solid–solution interface during the removal process [42]. The positive Δ*H*° indicates that the adsorption is endothermic in nature and that it is more favorable at higher temperatures. This may be related to the CPX interaction with active sites on the adsorbent surface, such as functional groups, or increased intra-particle diffusion within pores. Similar results have been reported for the removal of tetracycline with biochar [27] and hydrochar [43], where the adsorption was found to be a spontaneous and endothermic process.

### 2.8. Adsorption Process

Figure 7 describes the probable interactions between the antibiotic and adsorbent. The results obtained by Boehm titration, IR analysis, and changes in CPX removal efficiency with pH suggest three main interactions between the adsorbent and CPX, namely, hydrogen bonds, electrostatic interactions, and π–π interaction.

The Boehm titration results (see Table 1) indicate the presence of oxygenated groups such as phenols, carboxylic acids, and quinones. This is supported by the FT-IR results for the adsorbent (Appendix A). The presence of phenolic groups in the adsorbent suggests possible hydrogen bonding with the amino, thiol, and carboxylate groups of CPX. Electrostatic interactions are generated between the carboxylates of the adsorbent and protonated amines on CPX. These interactions were evident in the FT-IR analysis (Appendix A), where a decrease in the band of the –OH group (phenols and carboxylic acids) was observed in the adsorbent after CPX adsorption. The above results agree with the removal efficiencies observed with the change in pH. At pH < 6 (removal efficiency of ~90%), electrostatic interactions between deprotonated carboxylic acid and the protonated amino groups of CPX were predominant. At pH > 8, both the CPX and adsorbent were negatively charged, which generated electrostatic repulsion and agrees with the CPX removal efficiency of around 20%. In this case, the low removal efficiency was due to hydrogen bonding of the phenol in the adsorbent with the sulfur, nitrogen, and oxygen of CPX.

Other authors have shown that the oxygenated groups of the adsorbent participate in adsorption processes [6]. For example, [2] suggested that the oxygen-containing functional groups of the adsorbent, including hydroxyl, carboxyl, and alkoxy groups, may participate in the adsorption of antibiotics by forming hydrogen bonds with the phenolic hydroxyl, carboxyl, and amino groups of doxycycline and ciprofloxacin. Other interactions that contribute to CPX adsorption include the π–π interactions generated between the aromatic ring of CPX and aromatic fraction of the adsorbent, as evidenced by the displacement of the peaks of the aromatic C=C bonds around 1600 cm^−1^ after CPX adsorption. Similar results were reported by [5], who demonstrated the role of π–π interactions by preparing adsorbents with varying aromatic contents and showing that those with a higher aromatic fraction had a higher adsorption capacity. The above interactions were used to analyze the effects between components present in real matrices on CPX adsorption.

### 2.9. Effect of Complex Matrices on CFX Adsorption

Real wastewater has several components that may interfere with the adsorption process of the target molecule. In this study, the adsorption of CPX on F_Zn_ from traditional matrices such as wastewater and urine was evaluated. For comparison, CPX removal from distilled water was also considered. Appendix A presents the characteristics of each matrix. As shown in Figure 8, after 60 min, 71% and 77% of CPX were removed from the simulated wastewater and urine, respectively, while 96% of CPX was removed from the distilled water.

As expected, the removal efficiency decreased for both complex matrices. The tested matrices were characterized by a high inorganic salt content. Under working pH conditions and considering the PZC, the adsorbent was negatively charged and favored cation adsorption. Hence, cations such as Na^+^, K^+^, Ca^2+,^ Mg^2+^, and CPX may have competed for the active sites of the adsorbent (i.e., deprotonated carboxylic acids) [12]. On the other hand, the differences between the wastewater and urine were because the former had more cations. Despite the reduced CPX adsorption in these complex matrices, a removal efficiency of around 70% is quite good for actual applications.

## 3. Materials and Methods

### 3.1. Reagents

All reagents used in this study were of analytical grade. CPX (C_16_H_17_N_3_O_4_S) was purchased from Syntofarma and was used to prepare the synthetic solutions. HCl, NaOH, ZnCl_2_, and H_3_PO_4_ were purchased from Sigma-Aldrich (St. Louis, MO, USA).

### 3.2. Biochar Preparation

Fiber (F) and shell (S) residues from palm oil production were collected from CENIPALMA (Villavicencio, Colombia), washed with distilled water to remove impurities, and dried in an oven at 110 °C for 3 h. F and S were activated with ZnCl_2_ as per the previously reported procedure [19,44]. The activated adsorbents were designated as F_Zn_ and S_Zn_, respectively. To evaluate the effect of the activating agent, F was alternatively impregnated with H_3_PO_4_, and this adsorbent was designated as F_HA_. After preparation, all adsorbents were rinsed several times to reach a pH of 7. They were then dried in an oven at 110 °C, sieved to obtain a particle size of <0.15 mm, and stored in a dry place before use.

### 3.3. Biochar Characterization

The morphological properties of the adsorbents were characterized by scanning electron microscopy (SEM; Oxford Instruments, Concord, MA, USA). The surface area and pore volume were determined by nitrogen adsorption at 77 K with an ASAP 2020 Micrometrics instrument (Norcross, GA, USA.) according to the Brunauer–Emmett–Teller (BET) method [45]. Surface functional groups were analyzed by Fourier transform infrared (FT-IR) spectroscopy according to the attenuated total reflectance (ATR) (PerkinElmer, Waltham, MA, USA). Boehm titrations were performed following a standardized method to quantify the acidic groups on the adsorbent surface [19]. Finally, the point of zero charge (PZC) was obtained with the solid addition method.

### 3.4. Batch Adsorption Experiments

HCl (0.1 M) and NaOH (0.1 M) solutions were used to adjust the pH as required during the experiments. Adsorption experiments were carried out in batch mode. First, 100 mL of various initial concentrations of CPX and a certain amount of an adsorbent were placed into an Erlenmeyer flask and stirred constantly at 200 rpm. The effects of parameters such as the pH (2–12) at room temperature (298 K), adsorption time (0–60 min), CPX concentration (10–70 mg L^−1^), and adsorbent dose (0.5–2 g L^−1^) on the adsorption capacity were investigated. The adsorbent with the best performance was determined according to the CPX removal efficiency (%), as given in Equation (10), and the amount of CPX adsorbed at time *t* was calculated with Equation (11):(10)$$CPX\;Removal\;\left(\%\right)=\frac{C_{0}-C_{t}}{C_{0}}\times 100\%$$
(11)$$q_{t}=\frac{C_{0}-C_{t}}{w}\times v$$
where $q_{t}$ is the adsorption capacity of the adsorbent (mg/g); $C_{0}$ and $C_{t}$ (mg/L) are the CPX concentrations in the solution initially and at time *t*, respectively; v is the solution volume (L); and w is the mass of the adsorbent (g). To examine the diffusion mechanism involved during the adsorption process, three kinetic models were considered, namely, the pseudo-first-order (PFO), pseudo-second-order (PSO), and intra-particle diffusion models. The data were also adjusted to the Langmuir and Freundlich isotherms. These models were applied to the best adsorbent according to Equation (10). The linear coefficient correlation (*R*^2^ values) and the normalized standard deviation (Δ*q*, %) were evaluated to select the appropriate kinetic and isotherm models that best describe the measured data, as given below:(12)$$\Delta q\;\left(\%\right)=100\sqrt{\frac{\sum {\left(\frac{q_{exp}-q_{cal}}{q_{exp}}\right)}^{2}}{N-1}}$$

To evaluate the effect of the temperature on CPX adsorption, the thermodynamic parameters were studied at 15, 25, and 35 °C, where 0.16 g of adsorbent was added to 100 mL of CPX solution with a concentration of 20 mg L^−1^. The effect of complex matrices was evaluated by adding 0.16 g of adsorbent to 100 mL of simulated municipal wastewater and urine (see composition in Appendix A) doped with 20 mg L^−1^ CPX.

After the batch adsorption experiments, the samples were centrifuged and filtered with Whatman paper (0.2 µm). The residual CPX concentration was determined by high-performance liquid chromatography–ultraviolet (HPLC-UV) equipped with a reverse phase column (C18 column; annulus of 4.6 mm and length of 150 mm) and methanol carrier phase (30–70%) at a detection wavelength of 263 nm.

## 4. Conclusions

This study was focused on the understanding of the adsorption process of cephalexin onto biochar produced from fiber residue from palm oil activated with ZnCl_2_. The main results are as follows:

F_Zn_ is an effective CPX adsorbent, even in complex matrices as wastewater and urine.

The data for CPX adsorption on the adsorbent F_Zn_ were better fitted to the Langmuir model, suggesting homogeneous and monolayer adsorption of the pharmaceutical on the adsorbent surface. 

This process followed a PSO kinetic model, where the adsorption capacity was assumed as proportional to the square of the number of unoccupied sites. The adsorption process appears to be dominated by chemical adsorption through electron sharing or transfer between F_Zn_ and CPX.

The positive Δ*H*° indicates that the adsorption is endothermic and that it is more favorable at higher temperatures. This may be related to the CPX interacting with active sites on the adsorbent surface such as functional groups or the increased intra-particle diffusion within pores.

Based on the effects of the pH and pharmaceutical structure on the removal efficiency, the most likely adsorption mechanisms appear to be electrostatic interactions and hydrogen bonding, although π–π interactions also appear to contribute.

Finally, F_Zn_ was demonstrated to be a good adsorbent for charged pharmaceuticals, but was less effective for neutral substances.

This work also contributes to identifying the fiber residue from palm as a potential adsorbent to be applied for purifying wastewater that contains trace antibiotics.

## Figures and Tables

**Figure 1 molecules-26-03340-f001:**
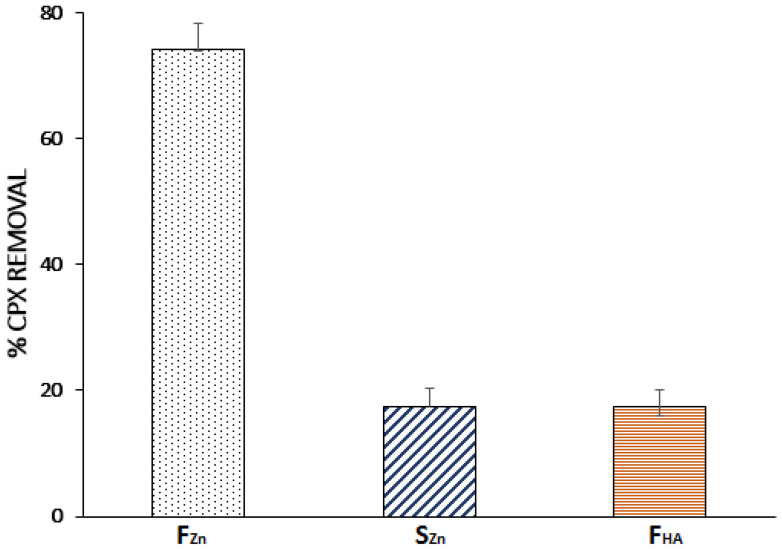
Performance of each adsorbent material for cephalexin (CPX) removal (adsorption conditions: (CPX): 20 mg L^−1^, adsorbent doses: 1.6 g L^−1^; time: 60 min, pH: 6.5).

**Figure 2 molecules-26-03340-f002:**
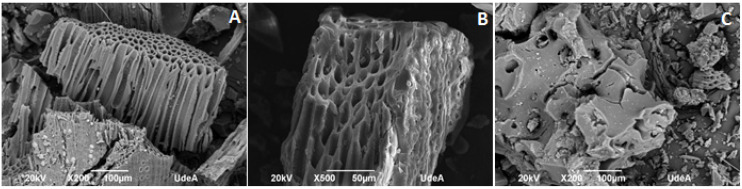
Scanning electron microscopy (SEM) micrographs of the adsorbent materials: (**A**) F_Zn_ (fiber activated with ZnCl_2_); (**B**) S_Zn_ (shell activated with ZnCl_2_); and (**C**) F_HA_ (fiber activated with phosphoric acid).

**Figure 3 molecules-26-03340-f003:**
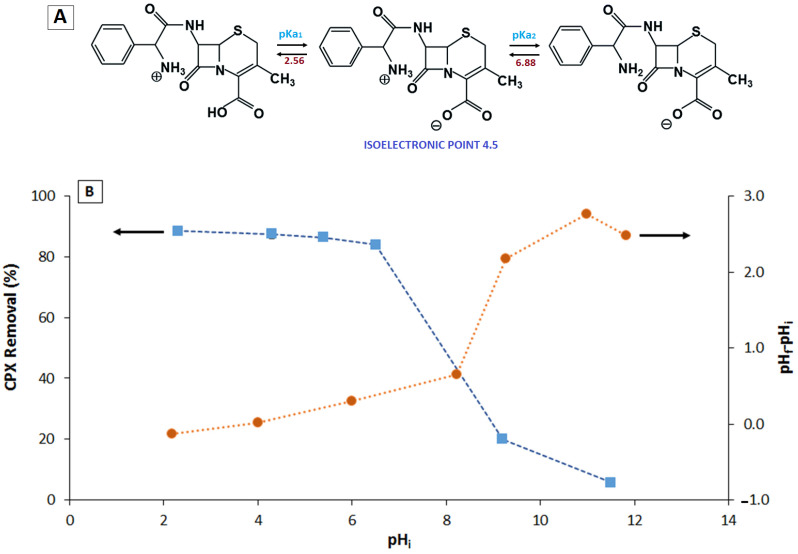
(**A**) Cephalexin ionization states and (**B**) (left) the effect of the solution pH on the CPX removal using F_Zn_ (adsorption conditions (CFX): 20 mg L^−1^, adsorbent doses: 1.6 g L^−1^, time: 60 min); (**B**) (right) point of zero charge (PZC) for the material.

**Figure 4 molecules-26-03340-f004:**
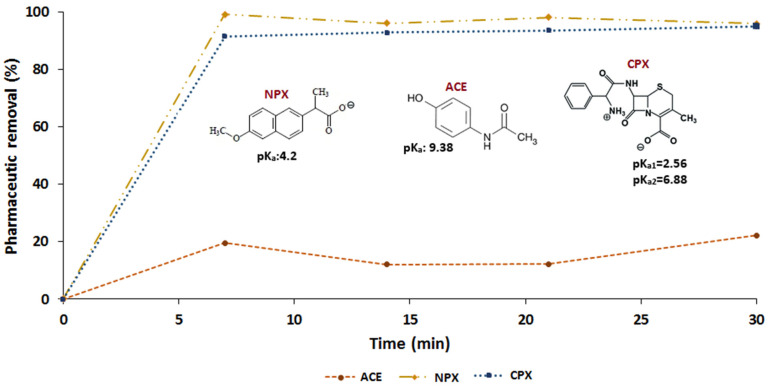
Adsorption of naproxen (NPX), acetaminophen (ACE), and cephalexin (CPX) on F_Zn_ material. Adsorption conditions (pharmaceutic): 20 mg L^−1^; adsorbent doses: 1.6 g L^−1^; pH: 4.0–5.0.

**Figure 5 molecules-26-03340-f005:**
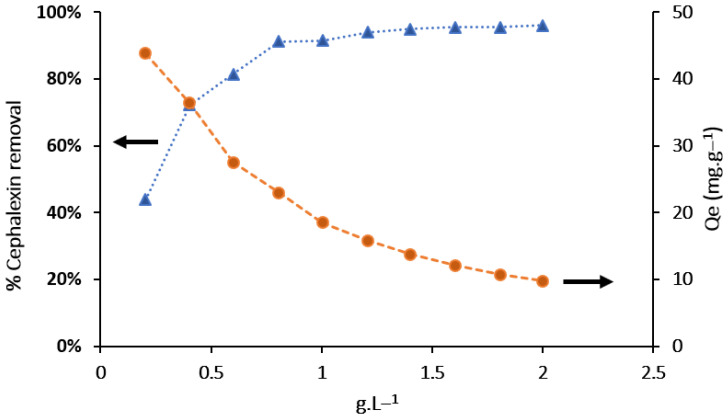
Effect of adsorbent dosage on the adsorption capacity and percentage removal of CPX (adsorption conditions: (CFX): 20 mg L^−1^, time: 180 min).

**Figure 6 molecules-26-03340-f006:**
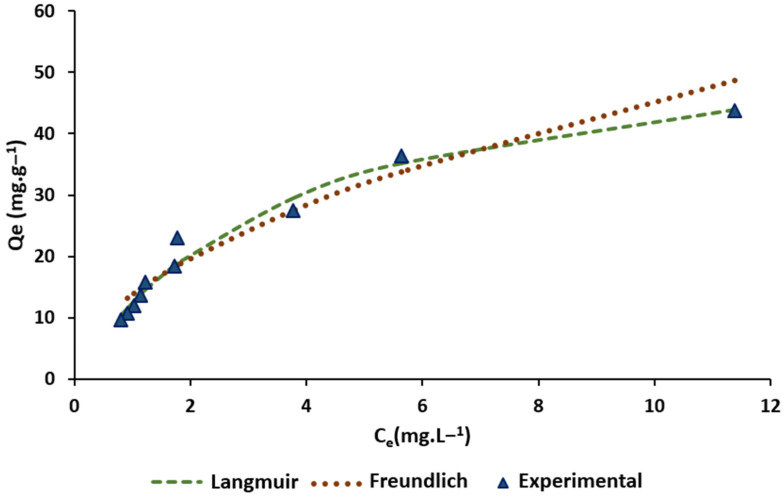
Experimental and theoretical isotherm values of CPX on F_Zn_ (180 min: adsorbent amount 0.02 to 0.2 g).

**Figure 7 molecules-26-03340-f007:**
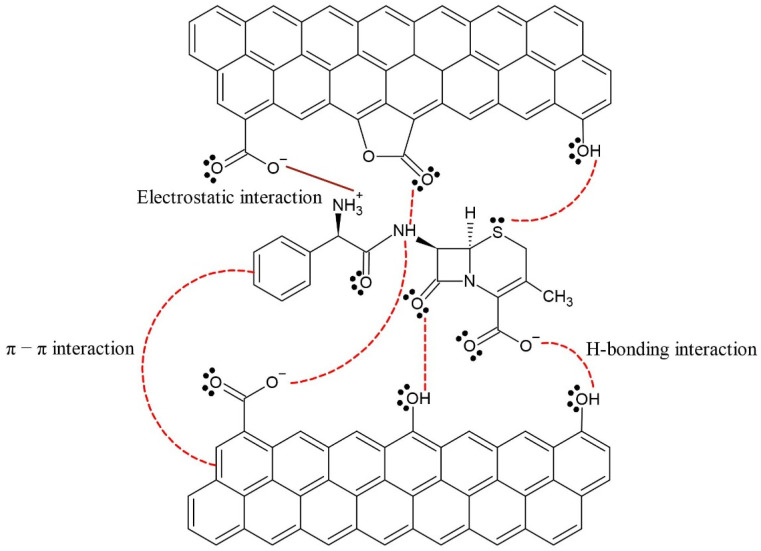
Possible interactions during CPX removal at pH: 4–5.

**Figure 8 molecules-26-03340-f008:**
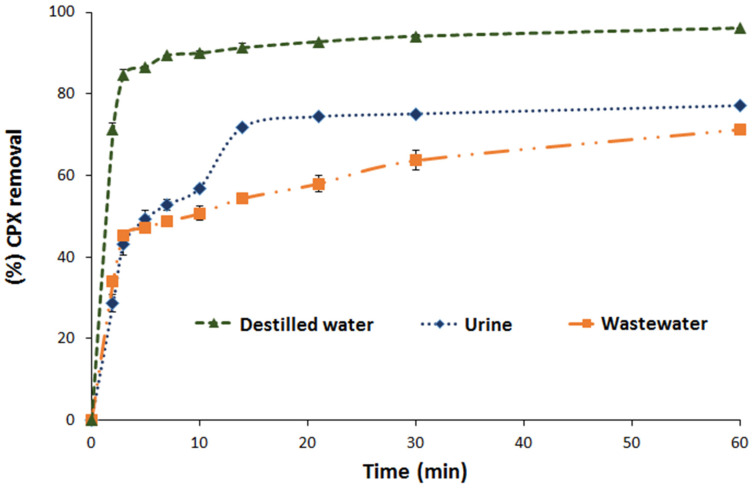
Adsorption of CPX in aqueous solution, in synthetic matrices of waste water and urine, using F_Zn_ (conditions: matrices doped with 25 mg L^−1^ of CPX, adsorbent dose 1.6 g L^−1^).

**Table 1 molecules-26-03340-t001:** Physical and chemical surface properties. PZC, point of zero charge.

Adsorbent Material	Surface Area (m^2^/g)	pH_PZC_	
F_Zn	835.3	4.0	
S_Zn	575.1	2.3	
F_HA	14.8	4.8	
Acid groups F_Zn	Carboxylic	Phenolic	Lactones
Value (mmol g^−1^)	1.65	0.20	0.15

**Table 2 molecules-26-03340-t002:** Adsorption kinetic parameters of CPX onto F_Zn_ (adsorbent doses: 1.6 g L^−1^, pH: 5 to 6; contact time: 60 min).

Initial Concentration							
	10 mg L^−1^	15 mg L^−1^	20 mg L^−1^	25 mg L^−1^	35 mg L^−1^	50 mg L^−1^	70 mg L^−1^
Experimental *q_e_* (mg g^−1^)	5.64	8.51	11.86	14.76	18.36	26.50	37.68
Pseudo-First Order							
*q_e_* (mg g^−1^)	------	0.59	1.12	1.68	4.68	6.18	3.39
*k*_1_ (min^−1^)	------	0.03	0.04	0.04	0.07	0.07	0.04
Δ*q_e_* (%)	------	93	91	89	74	77	91
*R* ^2^	------	0.969	0.999	0.990	0.989	0.928	0.889
Pseudo-Second Order							
*q_e_* (mg g^−1^)	5.68	8.56	12.0	14.8	18.9	27.3	38.17
*k*_2_ (min^−1^)	0.19	0.15	0.09	0.07	0.03	0.02	0.28
Δ*q_e_* (%)	0.59	0.54	1.56	0.24	2.8	2.8	1.28
*R* ^2^	0.999	1	1	0.999	1	0.999	0.999
Intraparticle Diffusion Model							
*k_d_* _1_	0.146	0.146	0.164	0.224	1.073	1.351	0.7947
*C* _1_	4.756	7.537	10.68	12.94	12.56	18.52	32.45
*R* ^2^	0.981	0.979	0.999	0.942	0.988	0.931	0.9751
*k_d_* _2_	0.022	0.033	−0.008	0.055	0.297	0.428	0.455
*C* _2_	5.39	8.160	11.945	14.163	16.077	23.234	34.18
*R* ^2^	0.831	0.980	0.7913	0.999	0.9812	0.9583	0.993

**Table 3 molecules-26-03340-t003:** Comparison of maximum adsorption capacity of CPX using biomass-derived adsorbents.

Biochar	*Q_m_* (mg g^−1^)	Reference
Fiber Palm	57.47	This study
Alligator weed	45.00	[2]
pomegranate peel	87.18	[38]
Albizia lebbeck seed pods	118.08	[9]

**Table 4 molecules-26-03340-t004:** Thermodynamic parameters of CPX adsorption on palm fiber adsorbent.

Temperature (°C)	Activation Energy, *E_a_* (kJ/mol)	Δ*H*° (kJ/mol)	Δ*S*° (J/mol)	Δ*G*° (kJ/mol)
15	91.6	62.2	228.5	−3.65
25				−5.93
30				−7.08

## Data Availability

Data are available from the authors on request.

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
