# Peer review of "Evaluating the Removal of the Antibiotic Cephalexin from Aqueous Solutions Using an Adsorbent Obtained from Palm Oil Fiber"

_molecules, 2021, doi:10.3390/molecules26113340_

Round 1

Reviewer 1 Report

As highlighted in the title, the scientific novelty of this work is the application of a biochar obtained from palm oil fiber for the removal of cephalexin from wastewater going in depth in the adsorption mechanism. The text is easy to follow, and the obtained results showed that the proposed methodology could be useful for environmental analysis applications. I think that the present work is within the scope of the journal and is remarkably interesting for other researchers in the field of environmental analysis.

However, in my opinion this work would need some corrections and could be accepted after major revision. Here some general questions and comments to address:

  1. Page 3 (line 99): As explain in section 2.4, 100 mL of sample were used for batch adsorption. Why was this volume selected? What about higher sample volumes? If the aim is applying this sorbent for water purification, it would be interesting to be able to do the treatment of high-water volumes (as higher as possible) being more interesting from the point of view of real application in a wastewater treatment plant.
  2. Page 3 (lines 118-120): Complex matrices were tested using a concentration of CPX of 20 mg/L. Did authors tested the application in wastewater and urine to check the matrix effect? Did authors consider a study of potential interferent species individually (e.g. cations or organic matter content)? In order to check the possible matrix it could be also interesting to do an interference study for each of the principal components of these kind of matrices at least at two concentrations (high and low).
  3. Page 3 (lines 130-131): Why a concentration of 20 mg/L CPX were selected for the further studies explained in section 3?
  4. Page 4 (line 144): To facilitate the comprehension of the explained results, it would be useful to include here the pka values of CPX.
  5. Pages 6-7, section 3.3: Naproxen and acetaminophen were also tested with the as-prepared FZn sorbent to check the influence of the ionizable functional groups of the target compound during extraction. However, what about the widely used amoxicillin and ibuprofen? Considering their chemical structure and ionizable functional groups both compounds can be also extracted using the sorbent proposed in this work. Did author consider an application of this sorbent for “multi-compound” extraction, for example cephalexin, amoxicillin, ibuprofen?
  6. Page 7, section 3.4: Using dispersive solid phase extraction strategy, there is an exhaustive extraction, not equilibrium. The amount of extracted CPX depends also on the sample volume. The maximum adsorption capacity would be determined by the amount of sorbent and as function of the sample volume.
  7. Figure 5 (caption): How Qe was calculated?
  8. Page 13 (lines 401-403): There is clearly a matrix effect. Did authors consider the use of complexing agents, such as EDTA (ethylenediaminetetraacetic acid is an aminopolycarboxylic acid), during extraction to reduce the negative effect of cations?

Reviewer 2 Report

This manuscript is good, but typical in the area investigated.

It is, I believe, rather long – including here the given, but well known equations; there is quite chattering in general.

Why “the best method for producing low-cost biochar adsorbents is through pyrolysis and subsequent chemical activation of the biomass waste” – & no ref. given ?

By “best” do the authors mean economically?

Perhaps, a hydrothermal process could be preferable {see, for instance, G.Z. Kyzas, E.A. Deliyanni, Modified activated carbons from potato peels as green environmental-friendly adsorbents for the treatment of pharmaceutical effluents, Chem. Eng. Res. Des., 97 (2015) 135–144}.

There are too many abbreviations (short names) used throughout being often confusing to the reader; some were unindentified, too (as PSO etc.).

I find the word “understanding” in the title a bit provocative - i.e. the conclusion:

“Based on the effects of the pH and pharmaceutical structure on the removal efficiency, the most likely adsorption mechanisms appear to be electrostatic interactions and hydrogen bonding, although π–π interactions also appear to contribute.”

I have also seen in the literature the following recent publication by the main author and her team:

Bioresource Technology 326 (2021) 124753, “Kinetics, isotherms, effect of structure, and computational analysis during the removal of three representative pharmaceuticals from water by adsorption using a biochar obtained from oil palm fiber”, and one of the studied pharmaceuticals is the current, CPX. Hence, the authors should refer to it for sure, meanwhile explaining what perhaps innovative exists in the present.

Round 2

Reviewer 1 Report

First, thanks to the authors for their work on the review. Responses to the comments are clear and concise.

After reviewing the changes and authors reponses, in my opinion the work can be accepted for publication in the present form. 

Reviewer 2 Report

As far as I am concerned and having seen the manuscript and the authors relative responses, the manuscript can be accepted in the current version.